# Experimental and Simulation Study of Micro-Channel Backplane Heat Pipe Air Conditioning System in Data Center

**Liping Zeng [1],\*, Xing Liu [1], Quan Zhang [2], Jun Yi [2], Xiaohua Li [1], Xianglong Liu [1] and Huan Su [1]**

[1] Hunan Engineering Research Center of Energy Saving and Material Technology of Green and Low Carbon Building, Hunan Institute of Engineering, Xiangtan 411104, China; edu_18163979051@163.com (X.L.); 13007449538@163.com (X.L.); 31014@hnie.edu.cn (X.L.); 17029@hnie.edu.cn (H.S.)

[2] College of Civil Engineering, Hunan University, Changsha 410082, China; quanzhang@hnu.edu.cn (Q.Z.); hezhiwen619@163.com (J.Y.)

\* Correspondence: zengliping223@163.com; Tel.: +86-180-0842-6267

**Abstract:** This paper mainly studies the heat transfer performance of backplane micro-channel heat pipes by establishing a steady-state numerical model. Compared with the experimental data, the heat transfer characteristics under different structure parameters and operating parameters were studied, and the change of heat transfer coefficient inside the system, the air outlet temperature of the back plate and the influence of different environmental factors on the heat transfer performance of the system were analyzed. The results show that the overall error between simulation results and experimental data is less than 10%. In the range of the optimal filling rate (FR = 64.40%–73.60%), the outlet temperature at the lowest point and the highest point of the evaporation section is 22.46 °C and 19.60 °C, the temperature difference does not exceed 3 °C, and the distribution gradient in vertical height is small and the air outlet temperature is uniform. The heat transfer coefficient between the evaporator and the condenser is larger than the heat transfer coefficient under the conditions of low and high liquid charge rate. It increases gradually along the flow direction, and decreases gradually with the flow rate of the condenser. When the width of the flat tube of the evaporator increases from 20 mm to 28 mm, the internal pressure drop of the evaporator decreases by 45.83% and the heat exchange increases by 18.34%. When the number of evaporator slices increases from 16 to 24, the heat transfer increases first and then decreases, with an overall decrease of 2.86% and an increase of 87.67% in the internal pressure drop of the evaporator. The inclination angle of the corrugation changes from 30° to 60°, and the heat transfer capacity and pressure drop increase. After the inclination angle is greater than 60°, the heat transfer capacity and resistance decrease. The results are of great significance to system optimization design and engineering practical application.

**Keywords:** micro-channel; backplane heat pipe; steady-state model; optimal filling rate; heat transfer characteristics

---

## 1. Introduction

With the rapid development of the Internet, the new generation of information technology, represented by big data, cloud computing, artificial intelligence and block chain, is accelerating the global intelligent transformation and promoting the substantial growth of data centers. According to a report issued by the Ministry of Industry and Information Technology in 2016, the market size of data centers in China has reached 71.45 billion yuan, and the power consumption of data centers accounts for 1.5% of the total power consumption of the whole society [1], and the power consumption of global data centers has also reached 1.1%~1.5% [2]. By 2020, data centers are expected to continue to grow [3],

and energy consumption is expected to increase along with demand. At the same time, reducing the energy consumption of high-heat density buildings in data centers has become a research hotspot. To reduce the energy consumption of high thermal density buildings, it is necessary to solve the heat dissipation of data center racks. At present, the data center usually adopts the method of separating the cold and hot channels to solve the problem of high thermal density. However, with the continuous increase of the heating density of the single frame, the high-temperature air outlet of the cabinet is easy to diffuse to the top of the frame, forming a short circuit of cold and hot air. When the method of isolating the cold and hot channels is adopted, the mixing of cold and hot air still exists. It has become a research point to improve the performance of the rack heat exchanger by adopting micro-channel structure heat pipes.

Micro-channel heat pipe heat exchangers have the advantages of both heat pipe and micro-channel structure, high heat transfer capacity, long transmission distance, simple processing technology. Wang [4] applied the separated heat pipe to the air conditioning system of the data center, introduced the application principle and working characteristics of the separated heat pipe in the air conditioning system of the data room, and analyzed its research status. Yan [5] studied this based on the gravity of the micro-channel flow boiling type start-up characteristics and heat transfer characteristics of the separated type heat pipe system, the experiment compared the two different channel lengths of aluminum micro-channel evaporator. The experimental results showed that in the circulation of the separated type heat pipe, micro-channel kernel boiling the bubble detachment diameter and the bubble frequency was the single value function of heat flux. Jia [6] comprehensively considered the influence of the inclination angle on heat transfer characteristics, flow characteristics, wear and ash accumulation characteristics in the evaporation section of the separated heat pipe exchanger, and determined the optimal design range of the inclination angle in the evaporation section was 7°~15°. Jin [7] took $CO_2$ as the refrigerant to study the influence of the liquid filling rate and the temperature difference between indoor and outdoor on the heat exchange and energy efficiency ratio of the system. The research showed that under the optimal liquid filling rate, with the increase of the temperature difference between indoor and outdoor, the heat exchange and energy efficiency ratio also gradually increased. Hu [8] used the micro-channel heat exchanger as the evaporator of the separated heat pipe, and compared it with the finned tube evaporator. The research showed that under the condition of equal heat exchange, the charging amount of the working medium in the micro-channel evaporator system decreased by 51.9%, while the system EER increased by 2.8%. Min [9] analyzed a heating system suitable for solar energy with a new micro-channel annular tube core part forming the influence of geometric parameters on the heat transfer capacity; the study found that the higher the effective porosity, the larger the aperture, and the bigger the fractal dimension of core samples, the higher the capillary limit; increases in the height difference between evaporator and condenser will also increase the heat transfer limits of the system. Ling [10–12] conducted experiments on the micro-channel separated heat pipe system of the base station and established a steady-state model. It was found that under different indoor conditions, the optimal filling rate (FR) remained at 88%~101%, and the system cooling capacity and EER increased with the increase of the indoor and outdoor temperature difference. Yuan [13] studied the micro-channel separated type heat pipe for experiments and simulations; it was concluded that the system was the optimal filling rate, and other working conditions of heat-exchange performance parameters and structure parameters and the influence of flow resistance, but the experimental value of mass flow and the simulation value of maximum relative error was 17.8%, therefore, to ignore the thermal resistance of the tube and tube inside and outside will have a direct impact on the heat exchange system, leading to a large data gap. Rezaei, et al. [14] numerically simulated the flow and heat transfer in the triangular section micro-channel, and studied the influence of the form and number of ribs on the heat transfer performance of the channel wall. The results showed that the rib had an effect on the physical properties of the flow, which was related to the Reynolds number. Jin [15,16] studied the micro-channel separated heat pipe with R134a as the medium, and the results showed that the temperature difference between 20 °C and 10 °C increased the heat

exchange by 106%, and the height difference between 1.2 m and 0.75 m increased by 267%. Khan [17] analyzed the relationship between liquid filling rate and heat transfer capacity, and compared the heat transfer capacity of each coolant. The experimental results showed that with the increase of filling ratio, the heat transfer capacity first increased and then basically remained unchanged. When the filling ratio was large enough, the heat transfer capacity decreased. Chen, et al. [18] analyzed the heat transfer characteristics of the heat pipe back panel air conditioning system, and the research showed that the back panel air conditioning can automatically adjust the heat exchange according to the changes in the environment. Sun et al. [19–21] proposed that the backplane heat pipe system has adaptability to the changes of different conditions, and the server should be placed in the middle and lower part as much as possible in order to play the maximum heat exchange capacity of the back plate. Compared with the traditional precision air conditioning, the anti-condensation backplane heat pipe system can save 20%–40% energy. Liu [22] analyzed the safety of the backplane heat pipe in the data center, and found that when a fan in the system was damaged, it had little impact on the heat transfer of the system, and the cabinet temperature rose 0.6~0.7 °C. Liu [23,24] carried out unbalanced operation analysis on the heat pipe back, and found the main causes of air conditioning system imbalance came from an uneven load of cabinet, in the same system, in the same system, do not retrieve more air-conditioning refrigerant, otherwise the upper part of the evaporator side is seriously dry, but it has little effect on the overall cabinet air intake temperature. Luo et al. [25] studied the extreme working conditions of the heat pipe back in the data room, and found that during normal operation, the temperature in the room was uniform. When part of the heat pipe back was malfunctioning, it was compensated by increasing the air volume to keep the inlet air temperature stable. Ding et al. [26] measured and analyzed the application of the heat pipe back plate in the data center, and conducted application studies in summer, transition season, and winter, respectively. The results showed that the heat pipe back plate could utilize natural cold source in transition season and winter, and the PUE in summer, winter and transition season was 1.58, 1.20 and 1.38, respectively. To sum up, although there are many researches on micro-channel heat pipes, there are few researches on the micro-channel backplane heat pipe system at present. Studies on the influence of internal environment changes on heat transfer performance of the micro-channel backplane heat pipe system under different structure parameters and operating parameters are lacking.

This paper proposes a micro-channel backplane heat pipe system. Unlike the traditional separated heat pipe, the evaporation end of the system is a backplane that uses phase difference to perform phase change heat. The micro-channel backplane heat pipe system is installed directly on the back of the rack. The fan extracts the high-temperature air from the side of the backplane evaporator and directly takes away the high-temperature heat inside the rack to achieve heat dissipation. The steady-state numerical model of the micro-channel backplane heat pipe system was established and the heat transfer performance of the system was analyzed by program simulation. First, R134a was used as the refrigerant, the heat transfer model of the system was programmed through MATLAB program operation and the result was compared with the experimental data. The comparative content includes the total heat transfer, import and export of evaporator and condenser parameters such as temperature and pressure, mass flow system, at the same time to simulate the different structural parameters and operation parameters of heat transfer, provide guidance for the application of practical engineering.

## 2. Experimental Method

### 2.1. Description of the Micro-Channel Backplane Heat Pipe Air Conditioning System

The backplane heat pipe air conditioning system is shown in Figure 1. The evaporator form of the backplane heat pipe system was the micro-channel heat pipe evaporator, and the condenser form was the copper brazed plate condenser CDU (Cooling Distribution Unit). The chilled water was provided by outdoor water chiller and cooling tower. The complete heat pipe backplane system consisted of back plate evaporator, gas pipe branch, CDU condensation section, liquid pipe branch

and other components. When the system was running, the refrigerant in the evaporation section of the back plate absorbed heat and became gaseous. The gaseous refrigerant rose through the gas pipe branch and entered the CDU condensation section. After heat exchange with the cooling water in the CDU, the gaseous refrigerant condensed into liquid. Under the effect of gravity, the refrigerant flowed down through the liquid pipe branch and returned to the evaporation section of the back plate. Such repeated circulation operation realizes heat transfer.

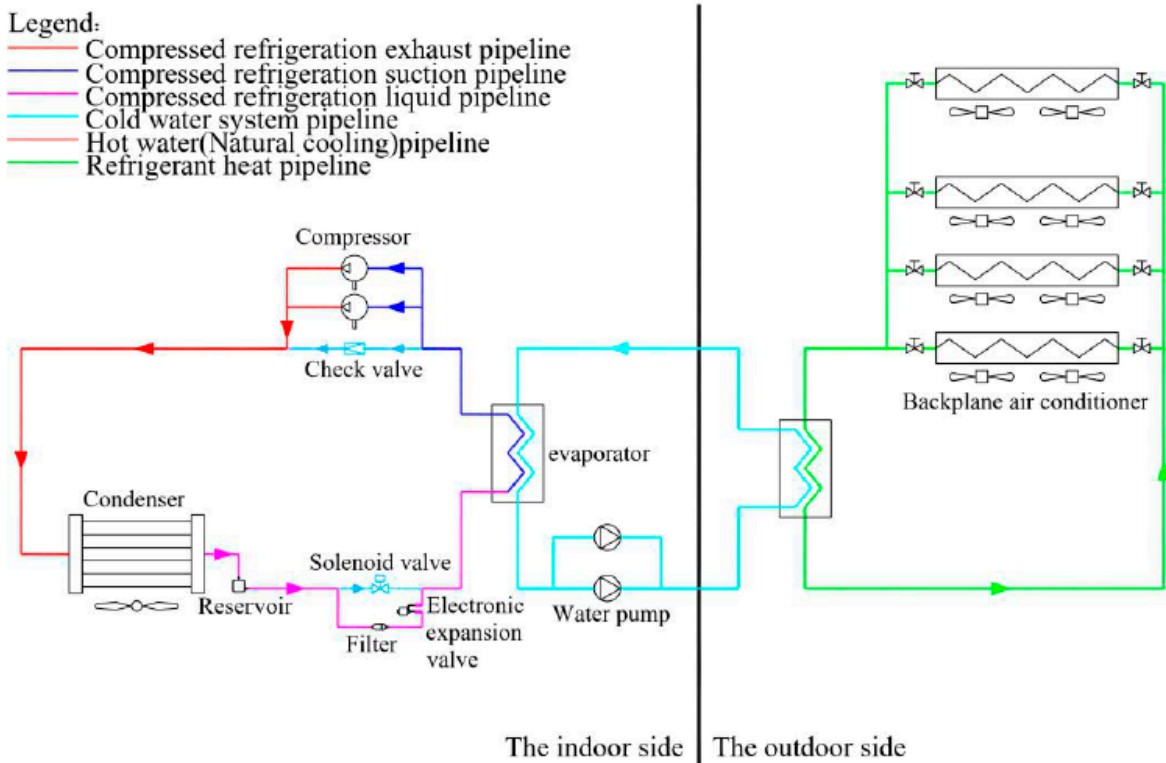

**Figure 1.** Backplane heat pipe air conditioning system.

## 2.2. Experimental Apparatus

The test unit included a back plate evaporator side (with cabinet), CDU condensation side, riser and downcomer. The experimental equipment included the enthalpy difference laboratory, which created the indoor side temperature and humidity environment conditions required by the backplane heat pipe experiment, opened and controlled the indoor water side equipment system, and provided the cold water supply temperature and flow required by the CDU cooling section experiment. The schematic diagram is shown in Figure 2 below. PT100 temperature sensor was used to measure the refrigerant temperature at the inlet and outlet of the backplane heat pipe evaporation section and CDU condensation section. K-type thermocouple temperature sensor was used to measure the wall temperature of the evaporation section of the backplane heat pipe. The pressure sensor was used to measure the refrigerant pressure at the inlet and outlet of the evaporation section and the condensation section of the backplane heat pipe. Coriolis flowmeter was used to measure the mass flow of refrigerant in the system. The experimental data measured by the experimental instrument was collected by Agilent data acquisition instrument and connected to a computer for storage.

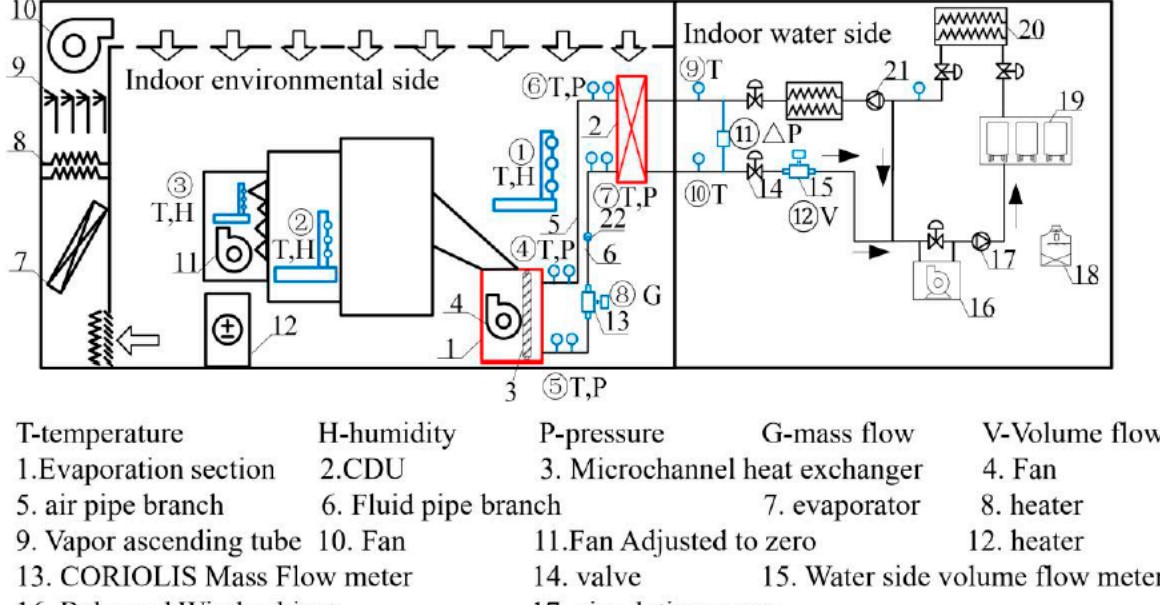

T-temperature     H-humidity     P-pressure     G-mass flow     V-Volume flow
1. Evaporation section     2. CDU     3. Microchannel heat exchanger     4. Fan
5. air pipe branch     6. Fluid pipe branch     7. evaporator     8. heater
9. Vapor ascending tube     10. Fan     11. Fan Adjusted to zero     12. heater
13. CORIOLIS Mass Flow meter     14. valve     15. Water side volume flow meter
16. Balanced Wind cabinet     17. circulating pump
18. Cooling tower     19. water chillers     20. Heating water tank
21. Variable frequency Water pump     22. sight glass

**Figure 2.** Schematic diagram of backplane heat pipe test platform of enthalpy difference laboratory.

*2.3. Experimental Procedures*

The standard operating conditions set in this experiment were: indoor dry/wet bulb temperature was 35 °C/24 °C, the backplane circulating air volume was 1800 m$^3$·h$^{-1}$, and cooling water. The supply/return water temperature was 12 °C/17 °C, and the flow rate was 1.71 m$^3$·h$^{-1}$. The experimental content was carried out according to the following three parts. The first part was to analyze the effect of different filling rates on the heat transfer performance of the system and determine the optimal filling rate under different conditions. The second part was the heat transfer performance analysis of the system after determining the optimal filling rate. The third part was the comparison of the heat transfer performance of different refrigerants. The effects of different temperatures, the air volume flow rate and other working conditions on the heat exchange performance of the system were analyzed. The other experimental conditions are set in Table 1. The experimental steps were the same as those of the standard working conditions. The influence of R22 and R134a refrigerants on the heat exchange of the system was analyzed.

**Table 1.** Other experimental conditions.

| Type | Numerical Value |
|---|---|
| Refrigerant charge (kg) | 0.6, 0.8, 1.0, 1.2, 1.4, 1.6, 1.8, 2.0 |
| Indoor temperature condition (dry/wet bulb temperature) (°C) | 28/18.9, 30/20.9, 35/24.9, 40/29.9 |
| Air volume (m$^3$/h) | 600, 800, 1000, 1200, 1400, 1600, 1800, 2000 |
| Chilled water inlet temperature (°C) | 10, 12, 14 |
| Chilled water flow (m$^3$/h) | 1.71 |
| Working substance | R22, R134a |

The standard working conditions of the experimental steps were as follows:

(1)   Through the enthalpy difference experiment control platform, the indoor environment temperature and humidity, water side water supply temperature and flow were set as the standard working conditions;

(2)   The measuring instrument was connected, the data acquisition instrument and the computer were turned on;

(3)   Refrigerant was added to the filling port, 0.6 kg for the first time, the enthalpy difference test bench was opened, the data recording interval of each group was set at 20 s, and the time was recorded for 30 min.

Step (3) was repeated to complete the experiment and data recording of other charging conditions.

## 3. Modeling

The numerical model of the micro-channel back plate heat pipe includes four parts: micro-channel evaporator, riser, CDU condenser and downcomer. The modeling follows the three laws of mass conservation, energy conservation and momentum conservation, and correspondingly calculates the refrigerant mass, heat exchange and pressure in each part, so as to establish a complete model of the system.

### 3.1. Steady State Heat Transfer Model of Evaporator

The evaporator model uses the finite volume method to divide each flat tube into several micro-element sections along the vertical height direction, and to calculate each micro-element section by section. In order to simplify the calculation, the following assumptions are made for the refrigerant in the flat tube:

(1)   The refrigerant flow in the flat tube is regarded as one-dimensional, and the axial heat transfer of refrigerant flow is not considered;

(2)   The refrigerant in each flat tube is evenly distributed, and the physical parameters of each refrigerant in the section in the flow direction and vertical direction are consistent;

(3)   The refrigerant in the collecting tube is evenly distributed. In the flat tube, the refrigerant at the inlet is super-cooled, the refrigerant in the middle is two-phase, and the refrigerant at the outlet is overheated. Different phase areas need to be calculated separately. The evaporator part of the back plate heat pipe system adopted in this paper is similar to that of the separated heat pipe in reference [27].

#### 3.1.1. Heat Transfer Model of the Refrigerant Side of Evaporator

(1) The single-phase area.

The single-phase area includes a super-cooled liquid zone and super-heated steam zone, and the calculation of the heat transfer coefficient adopts the correlation formula proposed by Gnielinski [28], and the formula is as follows:

$$Nu = \begin{cases} \frac{(f/8)(\text{Re}_{D_k}-1000)\text{Pr}}{1+12.7\sqrt{f/8}(\text{Pr}^{2/3}-1)}, & 2300 < \text{Re}_{D_k} < 10^6 \\ 4.6, & \text{Re}_{D_k} < 2300 \end{cases} \tag{1}$$

$$\text{Re}_{D_k} = \frac{G_r D_h}{\mu_r} \tag{2}$$

$$f = (1.82\log_{10}\text{Re}_{D_k} - 1.64)^{-2}. \tag{3}$$

The pressure drop is calculated as follows:

$$\Delta P = \Delta P_f + \Delta P_g = 2 \cdot f_1 \rho_r v_r^2 \frac{L}{D_t} + \rho_r g L \tag{4}$$

$$f_1 = \begin{cases} 16/\text{Re}_r & 0 < \text{Re}_r < 2500 \\ 0.079\text{Re}_r & 2500 < \text{Re}_r < 20000 \\ 0.46\text{Re}_r & \text{Re}_r > 20000 \end{cases} \tag{5}$$

(2) Two-phase area.

In the two-phase area, the correlation between Kim [29] is used to calculate the heat transfer coefficient. The calculation formula is as follows:

$$h_{tp} = \left( h^2{}_{nb} + h^2{}_{cb} \right)^{0.5} \tag{6}$$

$$h_{nb} = \left[ 2345 \left( Bo\frac{P_H}{P_F} \right)^{0.70} P_R{}^{0.38} (1-x)^{-0.51} \right] \left( 0.023\text{Re}_l{}^{0.8}\text{Pr}_l{}^{0.4}\frac{\lambda_l}{D_h} \right) \tag{7}$$

$$h_{cb} = \left[ 5.2 \left( Bo\frac{P_H}{P_F} \right)^{0.08} We_l{}^{-0.54} + 3.5 \left( \frac{1}{X_{tt}} \right)^{0.94} \left( \frac{\rho_g}{\rho_l} \right)^{0.25} \right] \left( 0.023\text{Re}_l{}^{0.8}\text{Pr}_l{}^{0.4}\frac{\lambda_l}{D_h} \right) \tag{8}$$

$$Bo = \frac{q_H''}{G_{h_{fg}}}, \ \text{Re}_l = \frac{G(1-x)D_h}{\mu_l}, \ We_l = \frac{G^2 D_h}{\rho_l \sigma}, \ X_{tt} = \left( \frac{\mu_l}{\mu_g} \right)^{0.1} \left( \frac{1-x}{x} \right)^{0.9} \left( \frac{\rho_g}{\rho_l} \right)^{0.5}. \tag{9}$$

The pressure drop in the two-phase area is calculated by Friedel [30] correlation, and the calculation formula is as follows:

$$\Delta P = \Delta P_f + \Delta P_g = \varphi_l{}^2 \frac{fL}{2D_h\rho_l}G^2 + \rho_m gL \tag{10}$$

$$\varphi_l{}^2 = E + \frac{3.24FX}{F_r{}^{0.045}We_l{}^{0.035}} \tag{11}$$

$$E = (1-x)^2 + x^2\frac{\rho_l\text{Re}_g{}^{-0.25}}{\rho_g\text{Re}_l{}^{-0.25}}, \ F = x^{0.78}(1-x)^{0.224} \tag{12}$$

$$X = \left( \frac{\rho_l}{\rho_g} \right)^{0.91} \left( \frac{\mu_g}{\mu_l} \right)^{0.19} \left( 1 - \frac{\mu_g}{\mu_l} \right)^{0.7}, \ F_r = \frac{G^2}{gD_h\rho_m{}^2} \tag{13}$$

$$f = \begin{cases} 16/\text{Re}_l & 0 < \text{Re}_l < 2500 \\ 0.079\text{Re}_l & 2500 < \text{Re}_l < 20000 \\ 0.46\text{Re}_l & \text{Re}_l > 20000 \end{cases}. \tag{14}$$

### 3.1.2. Heat Transfer Model of Air Side of Evaporator

The air side heat transfer coefficient is calculated by the J-factor correlation proposed by Kim Bullard [31], and the formula is as follows:

$$h_a = \frac{j\rho v c_{p,a}}{\text{Pr}_a{}^{2/3}} = \frac{j}{\text{Pr}_a{}^{2/3}}\frac{\rho v P_l}{\mu_a}\frac{c_{p,a}\mu_a}{\text{Pr}_a}\frac{\text{Pr}_a}{P_l} = j\text{Re}_a\text{Pr}_a{}^{1/3}\frac{\lambda_a}{P_l} \tag{15}$$

$$j = \text{Re}_a{}^{-0.487} \left( \frac{\theta}{90} \right)^{0.27} \left( \frac{P_f}{P_l} \right)^{-0.14} \left( \frac{H_f}{P_l} \right)^{-0.29} \left( \frac{B_f}{P_l} \right)^{-0.23} \left( \frac{L_l}{P_l} \right)^{0.68} \left( \frac{P_t}{P_l} \right)^{-0.28} \left( \frac{\delta_f}{P_l} \right)^{-0.05} \tag{16}$$

$$\text{Re}_a = \frac{G_a P_l}{A_{fe}\mu_a}. \tag{17}$$

### 3.2. Steady State Heat Transfer Model of Condenser

Similar to the evaporator, the condenser model uses the finite volume method to divide each plate into several micro-element sections along the vertical height direction, and calculates each micro-element section by section. In order to simplify the calculation, the following assumptions are made for the CDU plate condenser:

(1)　The flow of refrigerant and cold water between plates is regarded as one-dimensional flow, regardless of the axial heat transfer of refrigerant flow;

(2)　The refrigerant and cold water are evenly distributed in each flow channel, and the physical parameters of the refrigerant are the same on the cross-section of the flow direction and the vertical direction.

The plate has no heat capacity, and the temperature on both sides of the plate wall is the same.

### 3.2.1. Heat Transfer Model of the Refrigerant Side of Condenser

The basic heat transfer formula is used to calculate the heat exchange of each micro-element segment, and the schematic diagram of each micro-element segment is shown in Figure 3.

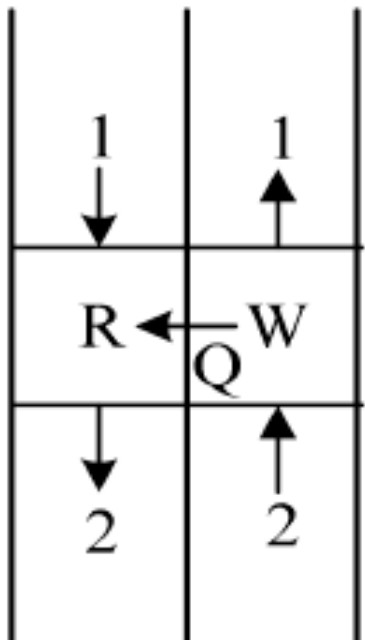

**Figure 3.** Schematic diagram of calculation micro-element segment.

For a given micro-element segment $j$, the heat exchange is calculated according to the following formula:

$$Q_{rc,j} = h_{rc}A_r(IT_{rc,j} - T_{wc,j}) = m_r(Ih_{r,in} - Oh_{r,out}). \tag{18}$$

Each calculation unit needs to calculate the heat exchange coefficient and temperature of the refrigerant first. The refrigerant temperature is calculated according to the pressure and enthalpy value at the outlet of the previous micro-element section. When the refrigerant is in a different phase state, the heat exchange coefficient and pressure calculation method are different.

According to the phase state of the refrigerant in the condenser, it can be divided into a superheated steam area, two-phase area and super-cooled liquid area. After judging the phase state of the refrigerant in the $j$th unit, the correlations between different heat transfer coefficients and pressure drop are used for calculation.

(1) The-single phase area.

The heat transfer coefficient of a single-phase region is calculated by the correlation formula proposed by Muley and Manglik [32]. The corrugation angle and area expansion coefficient are considered synthetically. The formula is as follows:

$$Nu_f = (0.2668 - 0.006967\beta + 7.244 \times 10^{-5}\beta^2)(20.78 - 50.94\varphi + 41.16\varphi^2 - 10.5\varphi^3)$$
$$\times Re_f^{0.728+0.0543\sin[(\pi\beta/45)+0.7]}Pr^{1/3}\left(\frac{\mu_f}{\mu_w}\right)^{0.14} \tag{19}$$

The pressure drop of refrigerant in the CDU plate condenser includes friction resistance, angle hole pressure drop and gravity pressure drop. The calculation formula of the single-phase friction pressure drop is as follows:

$$\Delta P_r = \Delta P_f + \Delta P_g \tag{20}$$

$$\Delta P_f = 2f\frac{L}{d_e}\rho\omega^2(\frac{\mu_f}{\mu_w})^{-0.17} \tag{21}$$

$$f = (2.917 - 0.1277\beta + 2.016 \times 10^{-3}\beta^2)(5.474 - 19.02\varphi + 18.93\varphi^2 - 5.34\varphi^3) \times Re_f^{-\{0.2 + 0.577\sin[(\pi\beta/45) + 2.1]\}} \tag{22}$$

$$\Delta P_G = \rho_j gL. \tag{23}$$

The calculation formula of refrigerant mass in the single phase area is as follows:

$$m_j = \rho_i A_c L_c \tag{24}$$

(2) The two phase area.

Yan [33] considered the influence of refrigerant heat flux, steam dryness and average heat flow, and proposed various plate heat exchangers suitable for R134a as refrigerant. Compared with the various condensation heat transfer coefficient formulas proposed earlier, they have a wider application range and are suitable for the establishment of this model.

$$Nu_f = 4.118Re_{eq}^{0.4}Pr_l^{1/3} \tag{25}$$

$$Re_{eq} = \frac{G_{eq}d_e}{\mu_l} \tag{26}$$

$$G_{eq} = G\left[1 - x + x(\frac{\rho_l}{\rho_g})^{0.5}\right]. \tag{27}$$

The calculation formula of friction pressure drop in the two-phase region of condensation heat exchange is as follows:

$$\Delta P_r = \Delta P_f + \Delta P_g \tag{28}$$

$$\Delta P_f = 2f\frac{L}{d_e}\rho\omega^2(\frac{\mu_f}{\mu_w})^{-0.17} \tag{29}$$

$$\Delta P_g = \rho_j gL \tag{30}$$

$$f = 94.75Re_{eq}^{-0.046}Re_l^{-0.4}Bo^{0.5}(\frac{p}{p_c})^{0.8} \tag{31}$$

$$Re_{eq} = \frac{G_{eq}d_e}{\mu_l} \tag{32}$$

$$Bo = \frac{qA_r}{m_r h_{fg}}. \tag{33}$$

The calculation formula of refrigerant mass in the two-phase area is as follows:

$$m_j = \rho_{tp,j} A_c L_c \tag{34}$$

$$\rho_{tp,j} = \rho_g \alpha + \rho_l(1 - \alpha) \tag{35}$$

$$\alpha = (1 + (\frac{1-x}{x})(\frac{\rho_g}{\rho_l})^{0.89}(\frac{\mu_g}{\mu_l})^{0.18})^{-1}. \tag{36}$$

### 3.2.2. Heat Transfer Model of the Cold Water Side of Condenser

The Muley and Manglik [32] correlation formula, which is the same as that in the single-phase region of the refrigerant, is used in the heat transfer formula on the water side, and the pressure drop on the water side is ignored and not calculated.

$$Q_{wc,j} = Q_{rc,j} = h_{wc}A_w(IT_{wc,j} - T_{wc,j}) = c_{p,w}m_w(T_{w,out,j} - T_{w,in,j}) \tag{37}$$

$$N\mu_w = (0.2668 - 0.006967\beta + 7.244 \times 10^{-5}\beta^2)(20.78 - 50.94\phi + 41.16\phi^2 - 10.5\phi^3)$$
$$\times Re_w^{0.728+0.0543\sin[(\pi\beta/45)+0.7]}Pr^{1/3}(\mu_l/\mu_w)^{0.14} \tag{38}$$

### 3.3. Steady State Heat Transfer Model of Connection Section

The connecting section of the back plate heat pipe system includes the rising gas pipe, the falling liquid pipe, the gas collecting pipe and the liquid collecting pipe in the evaporator and the condenser. These adiabatic sections are considered to have no heat exchange with the indoor environment, no heat loss, and the refrigerant flows in the adiabatic section with equal enthalpy, only pressure loss.

(1) Riser and downcomer.

The resistance of riser and downcomer mainly includes the local resistance and gravity resistance of the working fluid flow. The calculation formula of pressure drop is as follows:

$$\Delta P = \Delta P_c + \Delta P_g \tag{39}$$

$$\Delta P_c = (\lambda l + \sum \xi)\frac{\rho_r u_r^2}{2g} \tag{40}$$

$$\Delta P_g = \rho_r gH. \tag{41}$$

The refrigerant mass in the riser and downcomer:

$$M_{g1} = \rho_r A_{g1}L_{g1}. \tag{42}$$

(2) Gas collection pipe and liquid collection pipe.

There are many flow channels in the evaporator and condenser. During refrigerant distribution and collection, there is a pressure drop loss of sudden expansion and contraction. The pressure drop loss can be calculated by the following formula:

$$\Delta P_e = [(1-\delta)^2 - 0.4\delta]\frac{\rho V^2}{2} \tag{43}$$

$$\Delta P_C = (0.8 - 0.4\delta^2)\frac{\rho V^2}{2}. \tag{44}$$

The mass of refrigerant in the gas collecting pipe or liquid collecting pipe:

$$M_{g2} = \rho_r A_{g2}L_{g2}. \tag{45}$$

### 3.4. Programming Calculation of Steady-State Heat Transfer of the Micro-Channel Heat Pipe Backplane System

The operation of the whole system meets the three conservation laws of energy conservation, momentum conservation and mass conservation. Specifically, the temperature, pressure and enthalpy of the refrigerant in each component do not change after the completion of a cycle, which is equal to the value before the completion of the cycle. The mass flow of the refrigerant is consistent in the system. The specific calculation steps and calculation flowchart Figure 4 are as follows:

(1)     Input the structural dimension parameters of each component of the system, input the given working condition conditions, including the charging capacity M0, the temperature and flow at the inlet of the backplane, and CDU.

(2)     Start the calculation from the evaporator, assuming that in the initial calculation, the inlet enthalpy value is H0, the inlet pressure P0 and the mass flow G0 of the evaporator.

(3)     Use the refrigerant flow sequence to calculate the heat exchange, pressure drop and refrigerant mass of evaporator, gas collector, rising gas pipe, condenser, liquid collector and falling liquid pipe, respectively, according to the component model. The refrigerant outlet pressure, temperature and enthalpy of each component are assigned to the next component as the inlet parameter, and the refrigerant enthalpy H1 and pressure at the outlet are calculated after calculating the last component falling liquid pipe Force P1 and total mass M1.

(4)     Compare refrigerant enthalpy H1, pressure P1 and initial assumed values H0 and P0. If accuracy requirements are met, continue to compare mass M1 and initial assumed value M0. If not, correct initial assumed value H0 and return to step (2) for calculation. If mass M1 and initial assumed value M0 do not meet accuracy requirements, correct initial assumed value P0 and return to step (2) for calculation.

(5)     Repeat steps (2), (3) and (4) until the error convergence conditions are met, and output the import and export parameters of each component.

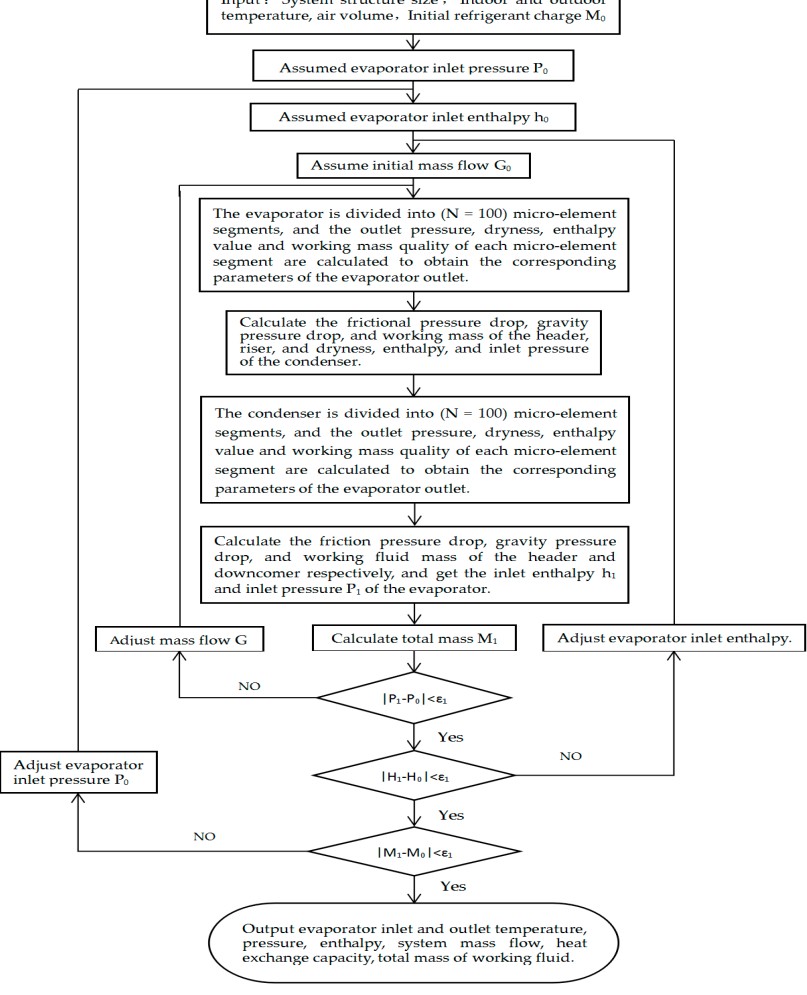

**Figure 4.** Flow chart of steady-state heat transfer calculation for the micro-channel backplane heat pipe system.

## 4. Result and Discussion

### 4.1. Verification of Model and Experimental Results

Figure 5 shows the comparison between the simulation value of heat exchange and the experimental value. It can be seen from the comparison that the error between the experimental value and the simulation value is large at low filling rate. When the filling rate range FR is greater than 40%, the simulation value is close to the experimental value, the relative error value is less than 10%, and when the filling rate FR is less than 40%, the relative error is more than 10%. Figure 6 shows the comparison between the simulated and experimental values of the average air temperature at the outlet of the evaporator side, which is similar to Figure 7. Figures 7–10 show the comparison between the simulated values and the experimental values of the refrigerant pressure and temperature at the inlet and outlet of the evaporator and the condenser. In the range of 0%~40% of the liquid filling rate, the error is large, and the maximum relative error is more than 20%. When the liquid filling rate is greater than 40%, the fitting result is good, and the average relative error is less than 10%. According to the analysis of all comparison results, when the filling rate is less than 40% (filling mass < 0.8 kg), the simulation results are relatively poor. On the one hand, the correlation between heat transfer and pressure drop is not accurate enough when the heat flow density is low, on the other hand, the heat pipe system itself cannot operate normally under the low filling rate, and the temperature and pressure sensors are not available when the operation state is not stable. Accurate temperature and pressure values were obtained by the method. In the actual operation, the system will be filled with refrigerant to the range of the best filling rate, so the model established in this paper is relatively accurate under the actual operation conditions, only for some of the more extreme operating conditions will there be deviation.

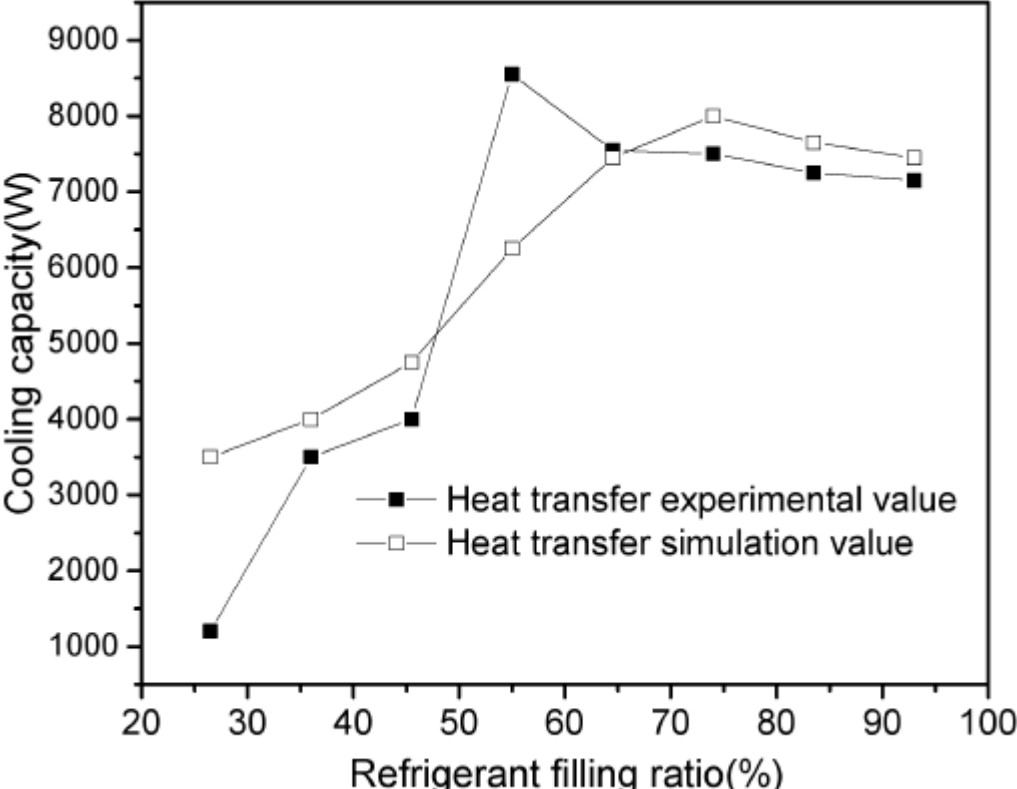

**Figure 5.** Comparison of experimental and simulated heat transfer values.

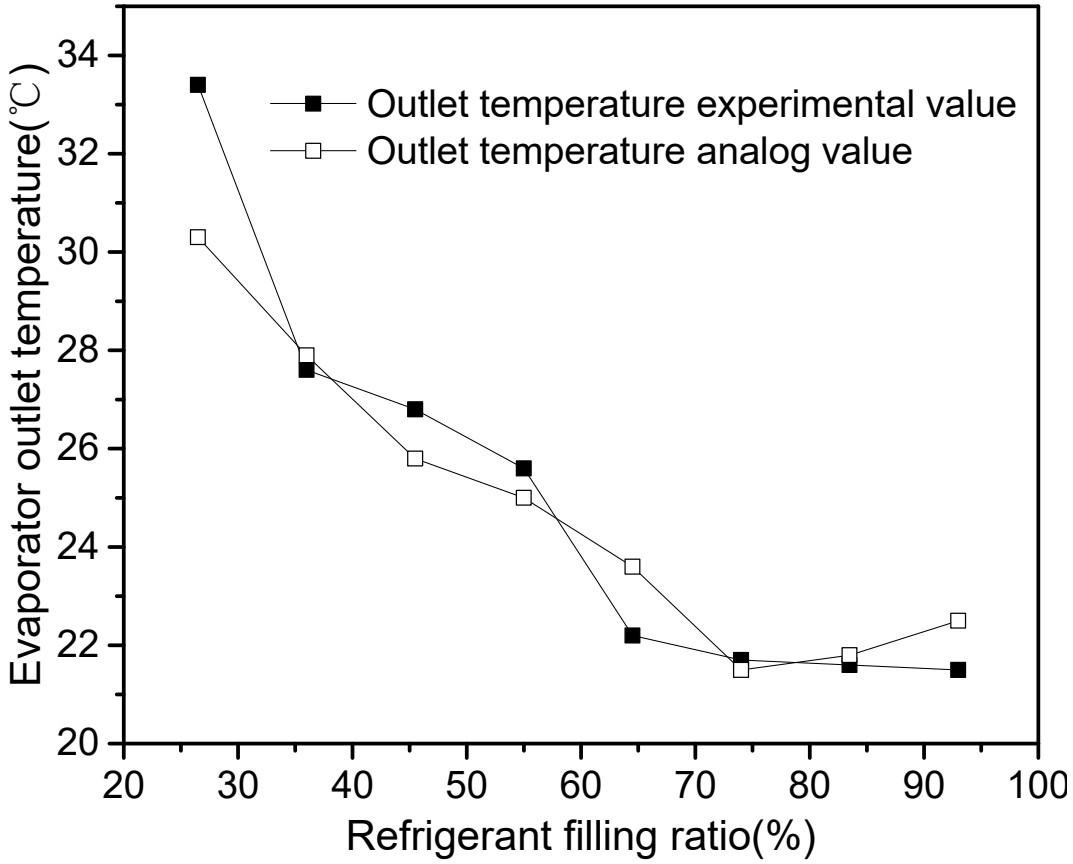

**Figure 6.** Comparison of air temperature at the outlet of the evaporator.

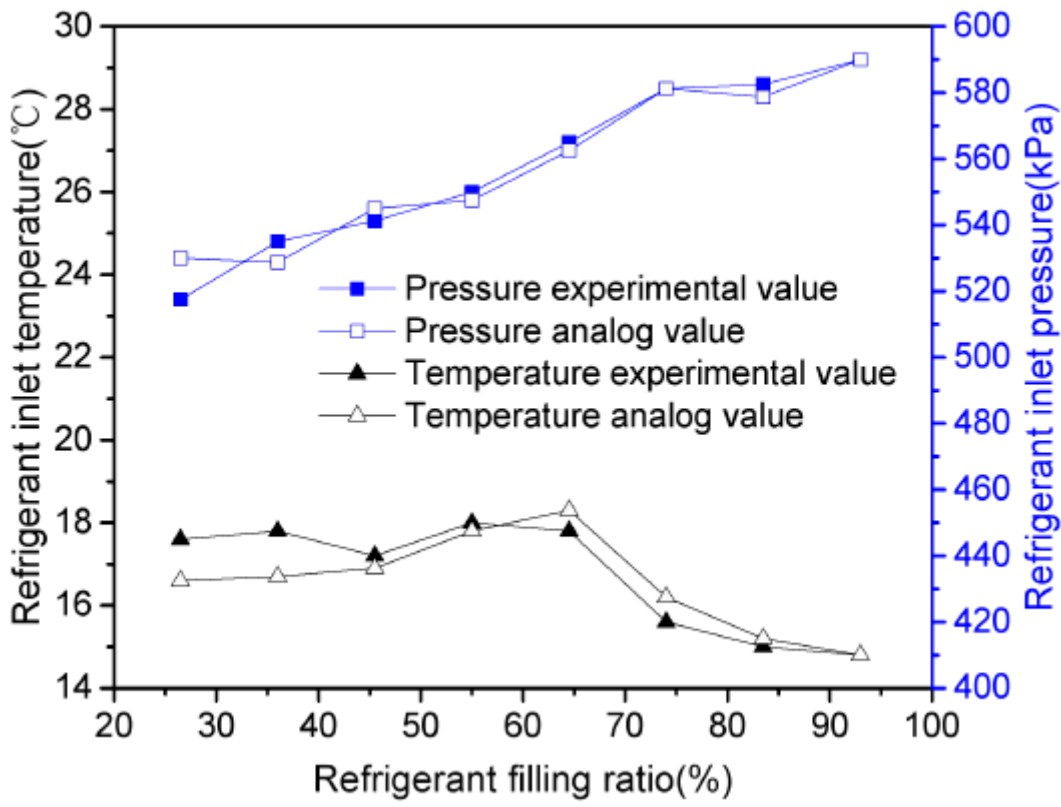

**Figure 7.** Comparison of temperature and pressure of refrigerant at the inlet of the evaporator.

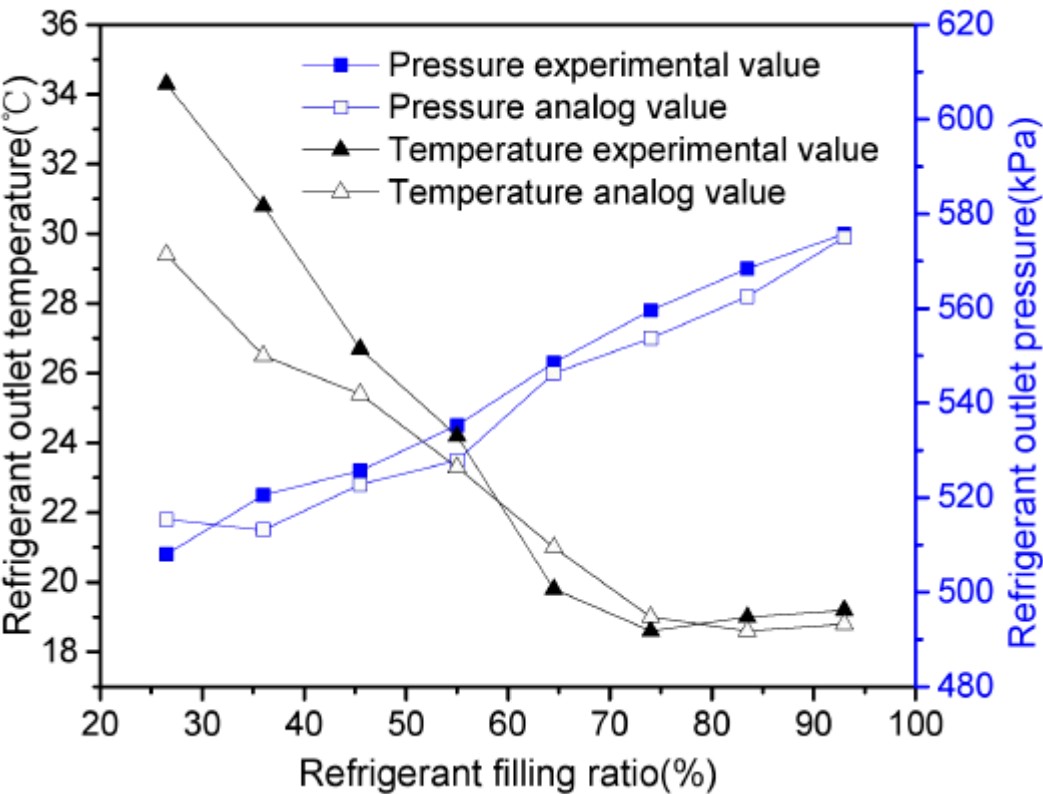

**Figure 8.** Comparison of temperature and pressure of refrigerant at the outlet of the evaporator.

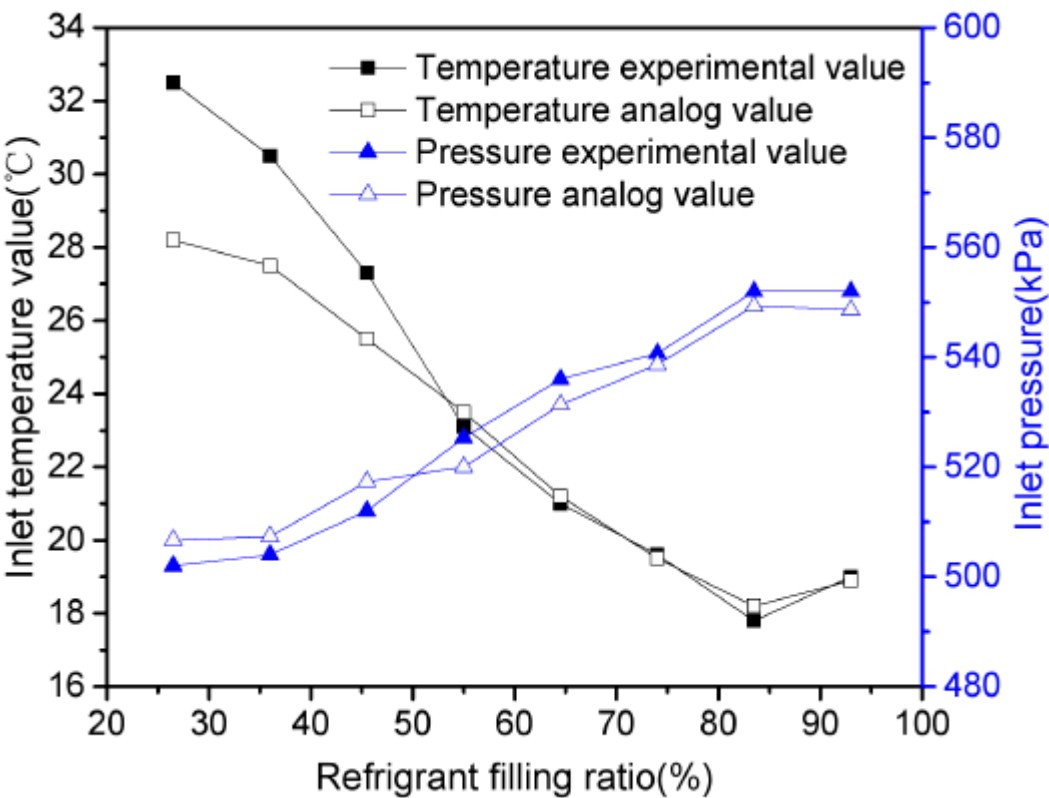

**Figure 9.** Temperature and pressure comparison of cooling distribution unit (CDU) inlet refrigerant.

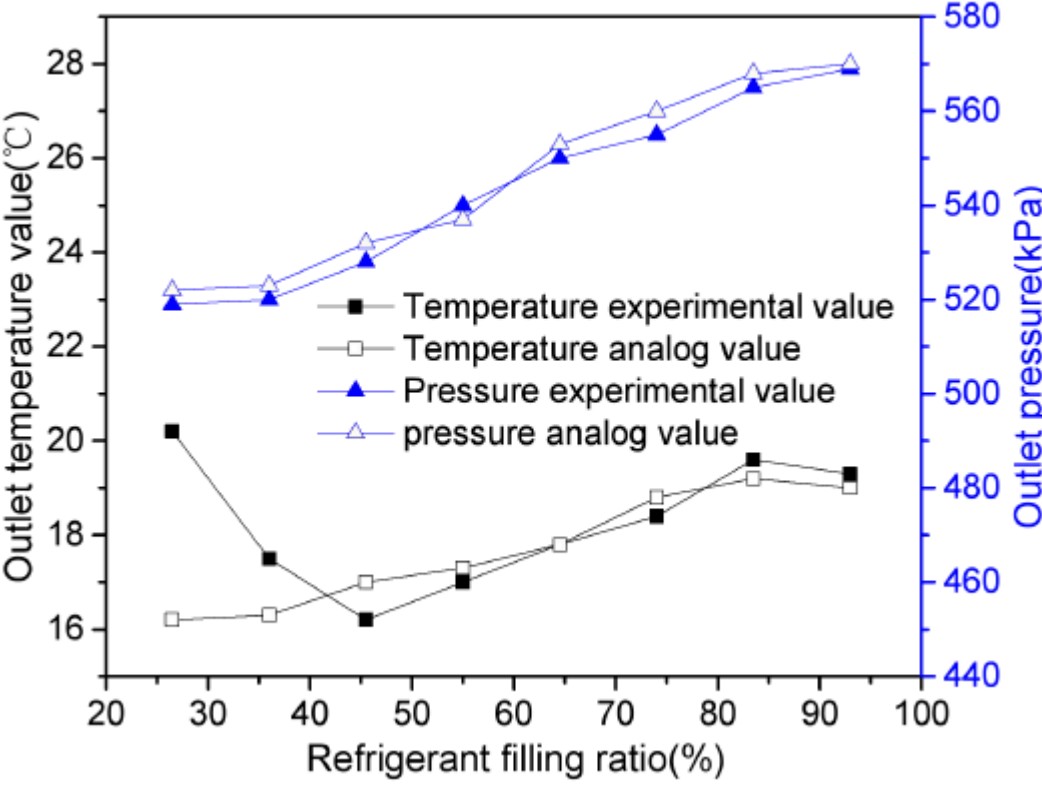

**Figure 10.** Temperature and pressure comparison of CDU outlet refrigerant.

This section is divided by subheadings. It provides a concise and precise description of the experimental results, their interpretation, as well as the experimental conclusions that can be drawn.

*4.2. Refrigerant Side Heat Transfer Coefficient Changes along the Process*

The distribution of refrigerant heat transfer coefficient in the evaporator along the calculation unit is shown in Figure 11. At low liquid filling rate (FR = 27.60%), the heat transfer coefficient in the evaporator kept decreasing, and at about the 50th stage, the heat transfer coefficient started to become very small because the refrigerant had completely turned into superheated steam before it reached the outlet in the evaporator; while at the optimal liquid filling rate range of FR = 64.40%–73.60%, the refrigerant in the evaporator was always in two phases The heat transfer coefficient was getting larger and larger, which was in the best working condition; when the refrigerant filling rate FR = 92.01%, although the heat transfer coefficient was increasing along the tube length, the overall heat transfer coefficient was significantly smaller than that of the best liquid filling rate. As can be seen from Figure 12, when the liquid filling rate was FR= 27.60%, the superheated steam at the outlet of the evaporator entered the condenser. In the condenser, the superheated steam first changed into saturated steam, and then the saturated steam released heat to form a liquid film. Therefore, the heat transfer coefficient increases first, then decreases, and the refrigerant undergoes phase transformation around the 20th calculation unit, and then the heat transfer coefficient gradually decreases. With the increase of the liquid filling rate and mass flow rate of the working medium, liquid accumulates in the condenser. When the liquid filling rate FR > 73.60%, the 75th unit near the outlet of the condenser appears super-cooling and the heat transfer coefficient decreases rapidly. When FR= 64.40%–73.60%, the working fluid is mainly in a two-phase state in the condenser, the heat transfer coefficient is significantly higher than other ranges, the decrease range of heat transfer coefficient is slower, and the heat transfer performance of the condenser is the best.

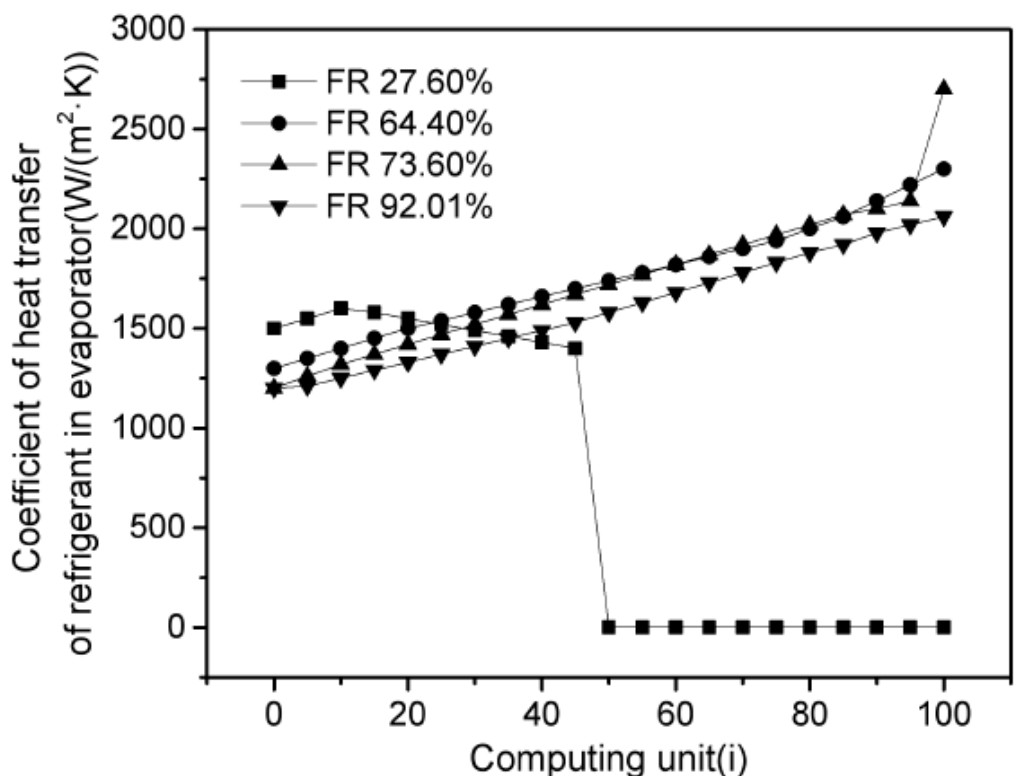

**Figure 11.** Schematic diagram of refrigerant heat transfer coefficient distribution along the calculation unit in the evaporator.

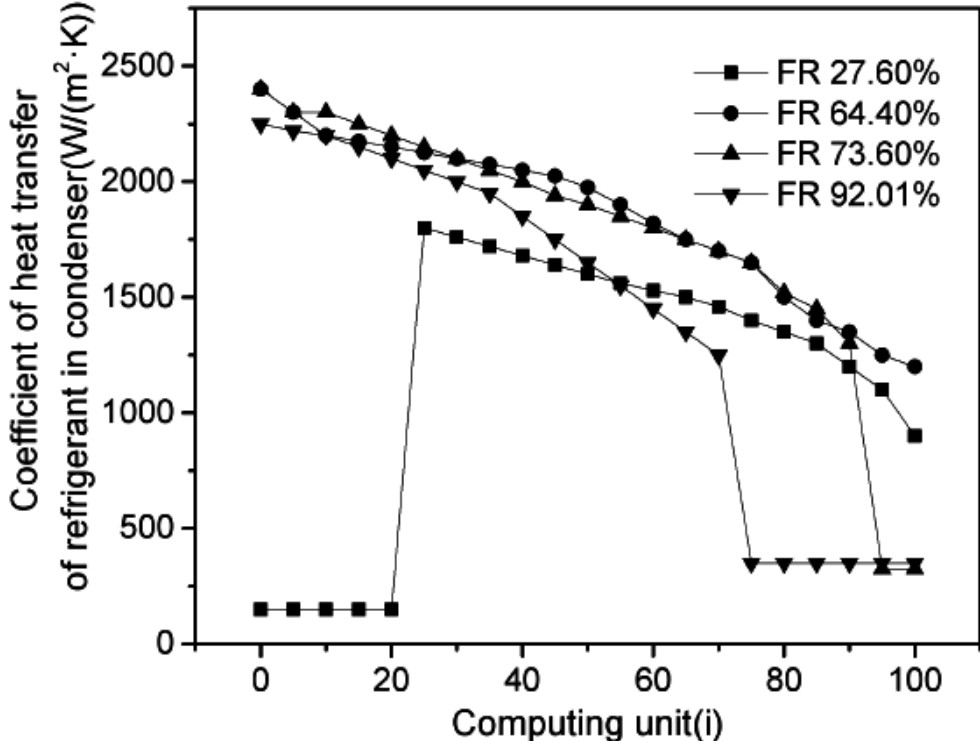

**Figure 12.** Schematic diagram of refrigerant heat transfer coefficient distribution along the calculation unit in the condenser.

### 4.3. Vertical Distribution of Back Panel Air Temperature

In order to get closer to the heat source of the cabinet, the evaporator in the micro-channel heat pipe back plate system was placed vertically, and the height of the evaporator was 158 mm. Because of the particularity of the structure of the back plate evaporator, the air outlet temperature of the back plate was different in the vertical height of the evaporator. In order to analyze the distribution of the air outlet temperature of the back plate, the program was used to calculate the change of the air outlet temperature of the back plate evaporator along the length of the evaporation section. The results are shown in the figure below.

According to Figure 13, it can be seen that the change law of the air side outlet temperature was consistent with the change law of the refrigerant side heat transfer coefficient in the evaporator. Under the conditions of low filling rate FR = 27.60% and high filling rate FR = 92.01%, the overall air outlet temperature of the back plate was higher than FR = 64.40%–73.60%. When FR < 27.60%, the outlet air temperature on the back side was close to the inlet air temperature at a height of about 72 mm, the refrigerant dried up in the evaporation section, and the evaporation section after this point had almost no refrigeration capacity. In addition, under the optimal liquid filling rate, it can be seen that the air outlet temperature at the lowest and highest points of the evaporation section was 22.46 °C and 19.60 °C, respectively, and the temperature difference did not exceed 3 °C, indicating that under the optimal liquid filling rate, the distribution gradient of the air outlet temperature at the vertical height of the backplane evaporator was relatively small and the air outlet temperature was relatively uniform.

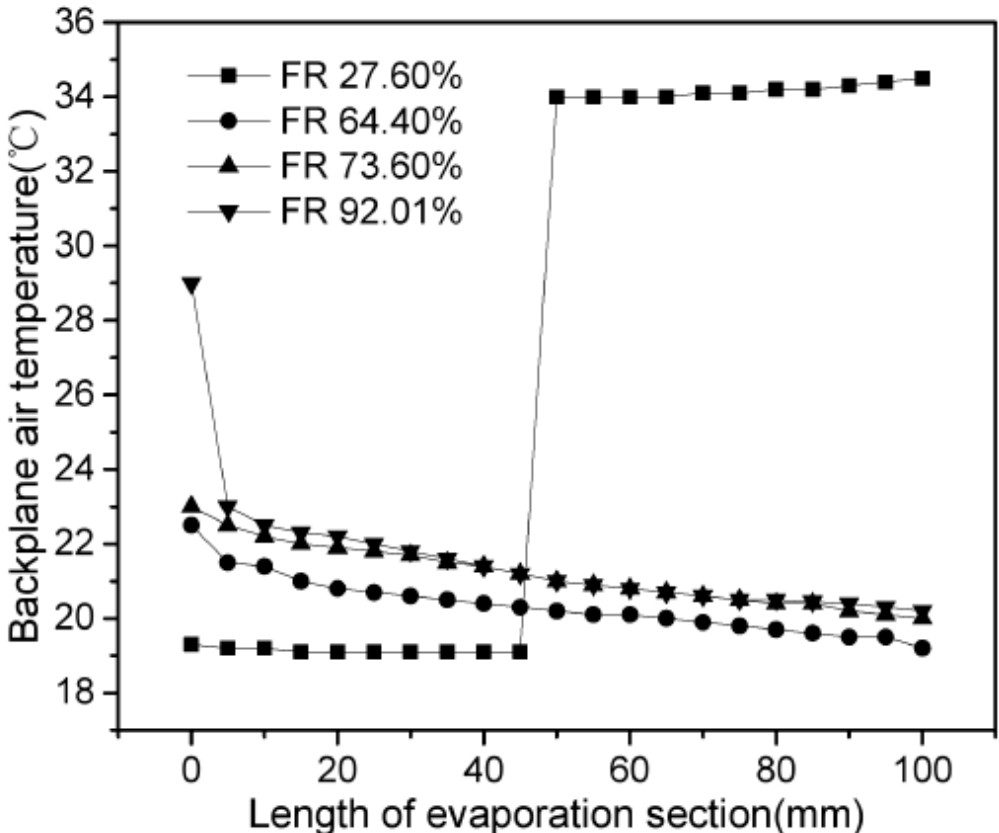

**Figure 13.** Variation of the temperature of the outlet air along the length of the evaporation section.

### 4.4. Influence of Structural Parameters of Heat Exchanger on Heat Transfer Performance

In the design of the heat exchanger, different structural parameters will have different effects on the performance of heat exchanger. For the heat exchange system of the back plate heat pipe, the structural parameters of the evaporator side of the backplane micro-channel and the CDU side of the

plate condenser have a great influence on the whole system. In this paper, starting from the evaporator side and the condenser side, the influence of the number of thin sections on the evaporator side and the width of the flat tube, the number of the condenser side plates, the ripple amplitude and the ripple inclination angle on the heat transfer performance are analyzed under the standard working conditions with different structure sizes.

### 4.4.1. Evaporator Side Structure

(1) Width of evaporator flat tube.

In this paper, the width of the flat tube in the evaporator structure was 24 mm, and the influence of different flat tube widths on heat transfer performance was simulated through the model. Other structural parameters were kept unchanged to increase the width of the flat tube from 20 mm to 28 mm. As can be seen from the Figure 14, the heat transfer increased with the increase of the width of the flat tube, and the increase gradually decreased. With the increase of the width of the flat tube, the direct heat exchange area between the refrigerant and the flat tube also increased, so the heat transfer gradually increased. The internal pressure drop of the evaporator decreased with the increase of the width of the flat tube. When the width of the flat tube increased from 20 mm to 28 mm, the internal pressure drop of the evaporator decreased from 26.51 kpa to 14.36 kPa. Because the effective flow area of the refrigerant increased, the hydraulic diameter increased, the refrigerant velocity decreased and the pressure drop decreased.

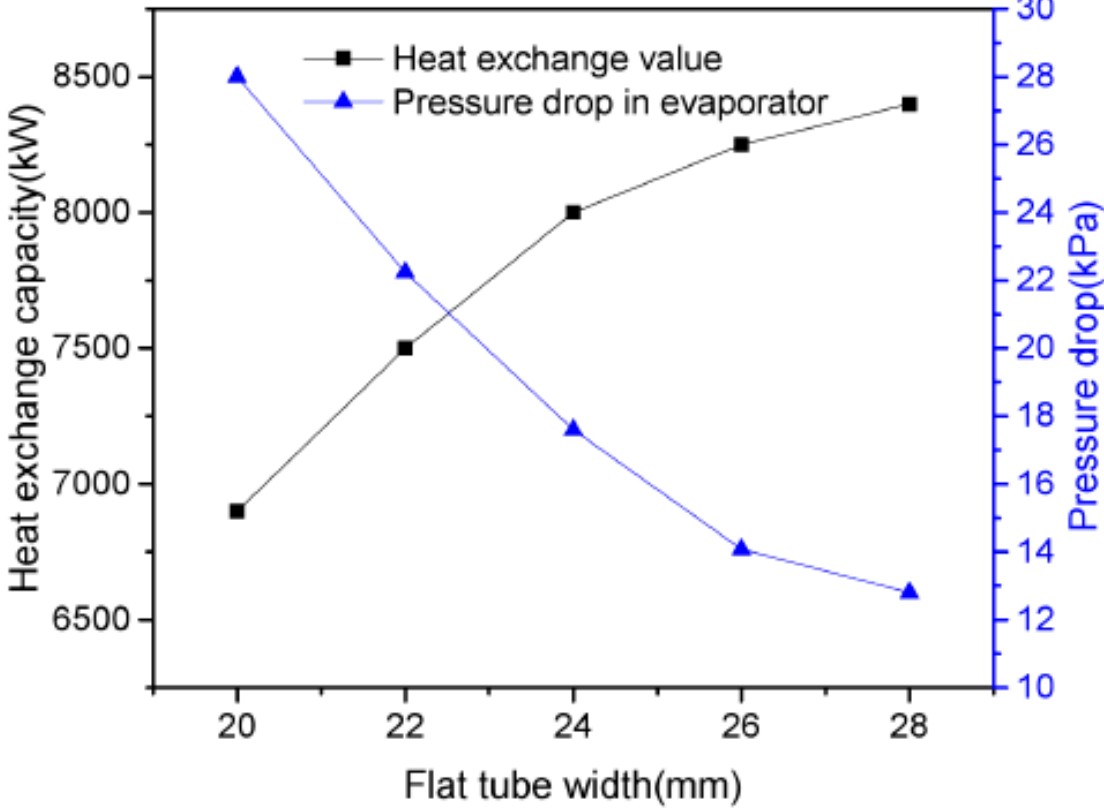

**Figure 14.** Heat exchange capacity and pressure drop in the evaporator as a function of flat tube width.

(2) Number of evaporator slices.

In evaporator micro-channel structure that includes a flat tube with multiple slices, the slices can increase the flat tube fluid disturbance, making the flat tube flow channel structure more compact, but the slices number also will increase the resistance of fluid in the flat tube. The research evaporator

structure adopted in the experiment included 20 slices, and the model simulated the effects of different numbers of the wafer heat-exchange performance.

As can be seen from the Figure 15, keeping other structural parameters unchanged, the number of slices was increased from 16 to 24 per flat tube. The chart indicates that while the number increased from 16 to 20, the heat was on the rise, and as slices continued to increase in number, the heat transfer fell sharply. As the slices number increased, the pressure drop in the evaporator increased. Because the flat tube slice number increased, refrigerant circulation area decreased; correspondingly, the hydraulic diameter was reduced, the refrigerant flow rate increased, the pressure drop increased.

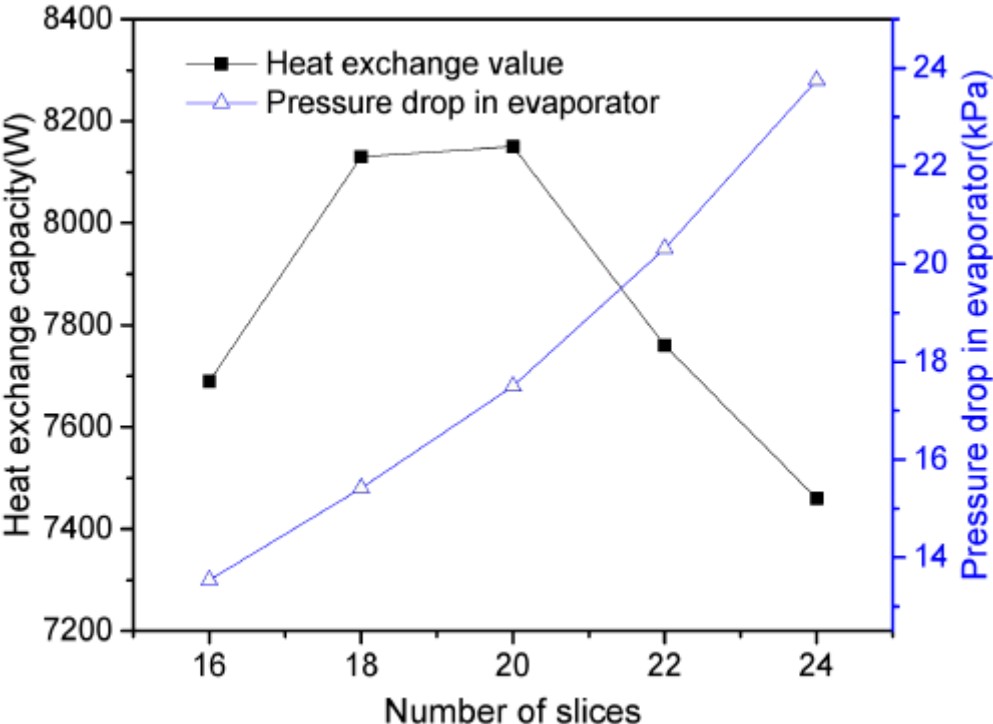

**Figure 15.** The heat exchange capacity and the pressure drop in the evaporator vary with the number of slices.

4.4.2. CDU Side Structure

(1) Number of CDU chips.

The condenser CDU used in the backplate heat pipe system was a brazed plate-type heat exchanger. The number of plates can be increased or decreased appropriately according to actual needs. The increase in the number of plates can increase the heat exchange area between the refrigerant and the frozen water side and increase the corresponding cost. The number of CDU condenser plates was 42, and the influence of a different number of plates on the heat transfer performance was simulated by the model.

Keeping other structural parameters unchanged, the number of condenser plates was increased from 22 to 62. As can be seen from Figure 16, both the heat transfer and pressure drop increased with the increase of the number of plates, and the increasing trend gradually decreased. As the number of plates increased, the heat exchange area increased, and the heat exchange correspondingly increased. Meanwhile, the frictional resistance of the refrigerant and the flow between the plates and the corner hole increased, and the pressure drop correspondingly increased.

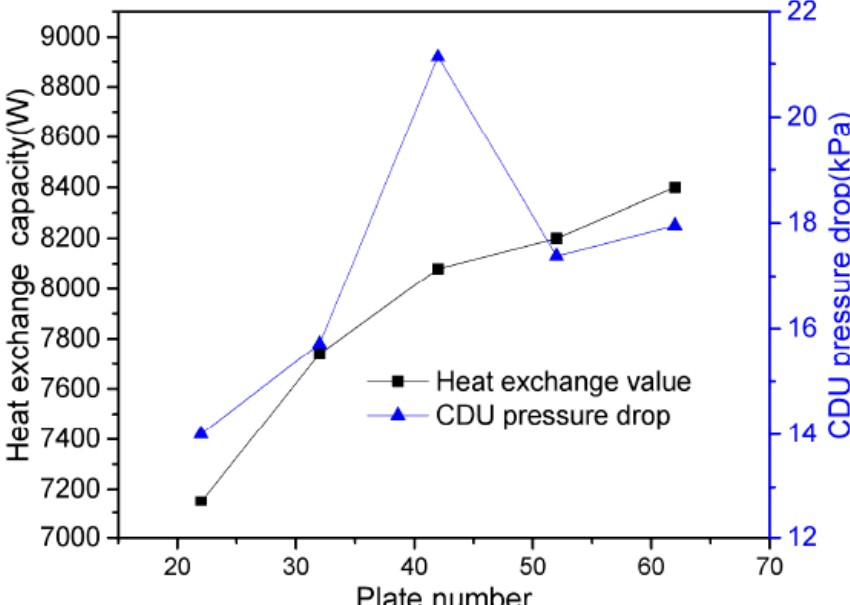

**Figure 16.** Heat exchange capacity and pressure drop in the evaporator vary with the number of plates.

(2) CDU ripple amplitude.

The amplitude of CDU ripple is the depth of ripple, and the size of ripple amplitude determines the flow form and heat transfer area of refrigerant between plates. The increase of ripple amplitude can enhance the mixing of vortices to some extent, but the possibility of scaling will also increase [34].

In this research, the ripple amplitude of CDU condenser was 24 mm, and the influence of different ripple amplitudes on heat transfer performance was simulated through the model. Other structural parameters were kept unchanged to increase the ripple amplitude of the condenser from 20 mm to 28 mm. As shown in Figure 17, the heat transfer increased with increasing ripple amplitude, refrigerant pressure dropped and decreased with the increment of ripple amplitude, among them, the variations in pressure drop was small, so the ripple amplitude of pressure dropped. Increased ripple amplitude is helpful to deepen the space between grooves, make the fluid turbulence degree strengthen, reduce the flow dead zone, and, correspondingly, the flow resistance is reduced, the pressure drop is reduced, and the heat transfer increased.

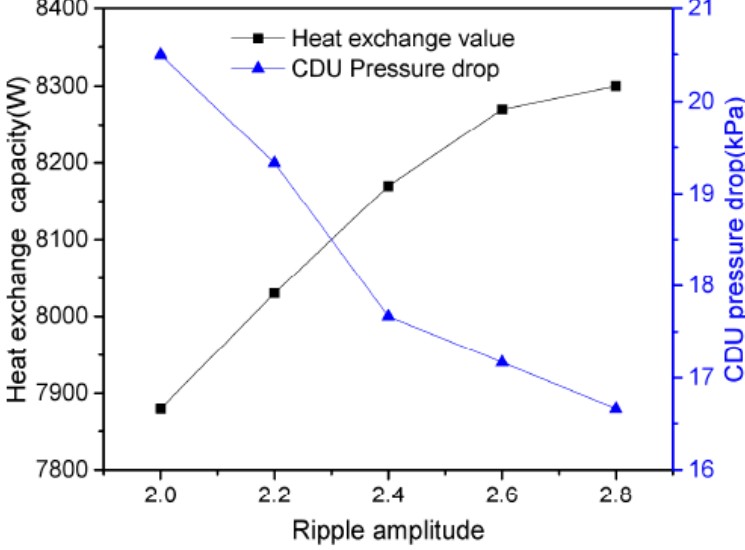

**Figure 17.** Heat transfer and pressure drop in the evaporator as a function of ripple amplitude.

(3) Ripple inclination of CDU.

When the inclination angle is small, the fluid flows along the direction of the ripple, and the distribution of the fluid is relatively uneven. When the inclination angle increases, the inter-plate fluid flows along the ripple decrease, forming a three-dimensional flow. Turbulence increases. In this paper, the refrigerant-side heat transfer correlation of the heat transfer model reflected the correction term for the ripple inclination angle, but it was still unable to completely simulate the effect of internal popular changes on temperature and pressure field, which had certain limitations. The ripple inclination angle of the CDU condenser was 30°, and the influence of different ripple inclination angles on heat transfer performance was simulated through the model.

Other structural parameters were kept unchanged to increase the corrugation angle of the condenser from 30° to 70°. As can be seen from the Figure 18, the ripple inclination angle changed from 30° to 60°, and the heat exchange and pressure drop increased. When the inclination angle was greater than 60°, the heat exchange and pressure decreased rapidly, which was the same as the trend of ripple inclination of a single plate simulated by Okada et al. [35]. When the inclination angle was greater than 60°, the inter-plate contact decreased, the effect of turbulence decreased, and the heat transfer and resistance decreased.

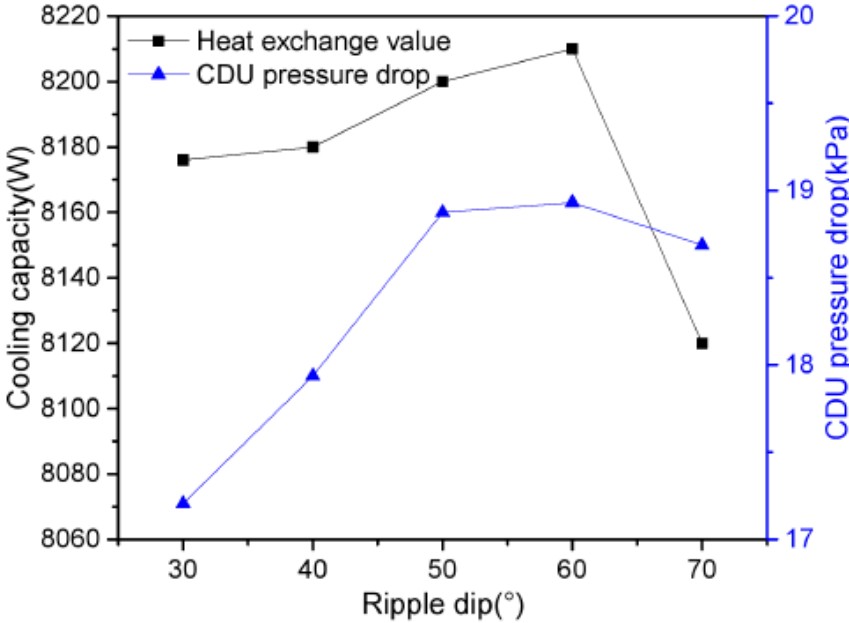

**Figure 18.** Variation of heat exchange capacity and pressure drop in the evaporator with the inclination of the corrugation.

## 5. Conclusions

In this paper, a micro-channel backplane heat pipe system was proposed to reduce the cooling energy consumption of the data center. Unlike the traditional separated heat pipe, the evaporation end of the system was a backplane that used phase difference to perform phase change heat. The micro-channel backplane heat pipe system was installed directly on the back of the rack. The fan extracted the high-temperature air from the side of the backplane evaporator and directly took away the high-temperature heat inside the rack to achieve heat dissipation. Based on the experiment, the system steady model was established, and further theoretically analyzed the influence of different parameters on the system to improve the heat exchange efficiency, reduce test times and experimental costs and provide guidance for the application of practical engineering and the following conclusions:

(1)    Using the data obtained from the experiment, compared with system model simulation results, the overall error is less than 10%.

(2) Under the conditions of low liquid filling rate, optimal liquid filling rate and high liquid filling rate, the heat transfer coefficient between the evaporator and the refrigerant side in the condenser was analyzed by the model. The refrigerant side heat transfer coefficient in the evaporator gradually increased along the flow, while the refrigerant side heat transfer coefficient in the condenser gradually decreased along the flow. The heat transfer coefficient between the evaporator and the condenser was higher than that under the conditions of low and high liquid filling rate.

(3) The model was used to analyze the distribution of the air outlet temperature of the back plate along the direction of the height. Under the condition of the optimal filling rate, the distribution gradient of the air outlet temperature of the back plate evaporator was relatively small in the vertical height, and the air outlet temperature was relatively uniform.

(4) The influence of structural parameter changes of the evaporator side and the condensing side on the overall heat transfer performance of the system was analyzed by using the model. When the width of the flat tube of the evaporator increased from 20 mm to 28 mm, the internal pressure drop of the evaporator decreased by 45.83% and the heat exchange increased by 18.34%. When the number of evaporator slices increased from 16 to 24, the heat transfer increased first and then decreased, with an overall decrease of 2.86% and an increase of 87.67% in the internal pressure drop of the evaporator. When the inclination angle of the ripple changed from 30° to 60°, the heat transfer and pressure drop increased. When the inclination angle was greater than 60°, the heat transfer and resistance decreased.

**Author Contributions:** Research: L.Z., X.L. (Xing Liu), Q.Z., X.L. (Xiaohua Li), J.Y., X.L. (Xiaolong Liu) and H.S.; Writing: L.Z., X.L. (Xing Liu), Q.Z., X.L. (Xiaohua Li), J.Y., X.L. (Xiaolong Liu) and H.S. All authors have read and agreed to the published version of the manuscript.

**Funding:** The research was funded by National Key R&D program of China grant number (2018YFE0111200) and (2016YFE0114300), Hunan provincial key projects grant number (2016JJ5011), The Provincial Natural Science Youth Foundation of Hunan grant number (2018JJ3102), Provincial Natural Science Foundation of Hunan, China, grant number (2018JJ2081) and (2018JJ4040), Scientific Research Fund of Hunan Provincial Education Department, China, grant number (17B064).

**Acknowledgments:** The authors are grateful for supported by construct program of applied specialty disciplines in Hunan province (Hunan Institute of Engineering).

**Conflicts of Interest:** The author declare no conflict of interest.

## Nomenclature

| | |
|---|---|
| Nu | refrigerant nusselt number |
| $Re_{Dh}$ | hydraulic diameter Reynolds number |
| $Re_l$ | liquid refrigerant Reynolds number |
| $Re_g$ | gas Reynolds number |
| $Re_a$ | Reynolds number on the air side |
| $R_{ef}$ | refrigerants Reynolds number |
| $R_{eq}$ | equivalent Reynolds Number |
| $Re_w$ | cold water Reynolds number |
| Pr | Prandtl number |
| $Pr_l$ | liquid refrigerant prandtl number |
| $Pr_a$ | air side prandtl number |
| $D_h$ | hydraulic diameter [m] |
| $G_r$ | refrigerant mass flux [kg/(m$^2$ × K)] |
| $H_{tp}$ | two-phase heat transfer coefficient [W/(m$^2$ × K)] |
| x | refrigerant dryness |

| | |
|---|---|
| $v_r$ | refrigerant flow rate [m $\times$ s$^{-1}$] |
| L | the cell length [m] |
| $\rho_r$ | density of refrigerant [(kg $\times$ m$^3$)$^{-1}$] |
| $A_{fe}$ | effective circulation cross-sectional area [m$^2$] |
| $A_s$ | windward area [m$^2$] |
| $\omega$ | ripple length |
| Ac | effective circulation area [m$^2$] |
| $G_{eq}$ | equivalent mass flow density [m$^3 \times$ s$^{-1}$] |
| $h_{fg}$ | latent heat of vaporization [J $\times$ kg$^{-1}$] |
| $m_j$ | refrigerant mass of unit j [kg] |
| $\rho_r$ | refrigerant density [(kg $\times$ m$^3$)$^{-1}$] |
| $\mu_1$ | saturated liquid viscosity [Pa $\times$ s] |
| $\rho_1$ | saturated liquid phase density [(kg $\times$ m$^3$)$^{-1}$] |
| $P_r$ | Pcrit refrigerant pressure [Pa] |
| $h_a$ | air side heat transfer coefficient [W $\times$ (m2 $\times$ K)$^{-1}$] |
| $I_{hr,in}$ | unit refrigerant inlet enthalpy [J $\times$ kg$^{-1}$] |
| $P_H, P_F$ | heated perimeter channel wet week [m] |
| $\rho_{tp,j}$ | refrigerant density in two-phase region [(kg $\times$ m$^3$)$^{-1}$] |
| $T_{w,in,j}$ | cold water inlet temperatures of the j th unit [K] |
| $A_g$ | cross section of the ascending or descending pipe [m$^2$] |
| H | height difference between evaporator and condenser [m] |
| $M_g$ | mass of refrigerant in the ascending or descending pipe [kg] |
| $\Lambda$ | liquid refrigerant coefficient of thermal conductivity [W $\times$ (m K)$^{-1}$] |
| $\Phi_{12}$ | friction multiplier |
| $\beta$ | ripple dip [°] |
| $m_r$ | refrigerant flow [kg $\times$ s$^{-1}$] |
| $\alpha$ | section gas content |
| E, F | enhancement factor and inhibitory factor |
| $m_w$ | cold water flow [kg $\times$ s$^{-1}$] |
| $IT_{wc,j}$ | cold water side temperature of j th unit [K] |
| $\lambda$ | along-path resistance coefficient |
| $\zeta$ | local resistance coefficient |
| $V/\mu_r$ | refrigerant flow rate [m$^3 \times$ s$^{-1}$] |
| $Q_{c,j}$ | unit refrigerant heat exchange capacity [W] |
| $L_{g2}$ | length of riser or descender [m] |
| $\delta$ | area ratio |
| $\mu_r$ | refrigerant dynamic viscosity [Pa $\times$ s] |
| $q''_H$ | effective heat flux density [W/m$^2$] |
| Bo | Bond number |
| $X_{tt}$ | Marty number |
| f | friction coefficient |
| $T_{wc,j}$ | unit refrigerant side wall temperature [K] |
| $A_r$ | micro-element heat exchange area [m$^2$] |
| B/b | ripple amplitude [m$^2$] |
| $\Phi$ | area to expand coefficient |
| G | actual mass flow [m$^3 \times$ s$^{-1}$] |
| Q | heat flux density [(W $\times$ m$^2$)] |
| $M_r$ | refrigerant mass flow [J $\times$ kg$^{-1}$] |
| $L_g$ | length of riser or descender [m] |
| $T_{rc,j}$ | unit refrigerant temperature [K] |
| $\mu_g$ | saturated vapor phase dynamic viscosity |

| $\rho_g$ | saturated gas phase density $[(kg \times m^3)^{-1}]$ |
| $P_{crit}$ | refrigerant critical pressure [Pa] |
| $T_{w,out,j}$ | outlet temperatures of the j th unit [K] |
| $O_{hr,out}$ | unit refrigerant outlet enthalpy $[J \times kg^{-1}]$ |
| EER | Cooling capacity (W)/input power (W). |
| PUE | Total facility power/IT equipment power. |

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
