# Peer review of "Experimental and Simulation Study of Micro-Channel Backplane Heat Pipe Air Conditioning System in Data Center"

_applsci, doi:10.3390/app10041255_

Round 1
Reviewer 1 Report
the paper is well written and organized. it should be improved by specifically mentioning its original contribution compared to the state of the art, both in the intro and in the conclusion, where the authors should specify how they met their purpose.
Author Response
Thanks the respectable reviewer for her/his valuable comment. We applied your kind comments to the manuscript to improve it. To specify, in the intro and conclusion part, we especially highlighted the improvement and advantage of the micro-channel back plane heat system. For the modified part, we marked with red font at the page 5, line 127-132 and the page 22, line 526-535.
Reviewer 2 Report
Dear authors
- You present research Experimental and simulation study of micro-channel backplane heat pipe air conditioning system in data center. In this study, is shown how there are many researc,hes on micro-chanel heat pipes, but there are few research on the micro-chanel backplane heat pipe system at present.
The post is very valuable, due to the power consumption of data centers accounts for 1.5% of total power consumption, and currently, we need to decrease CO2 emissions. The use of this energy would be important.
The document presents a good structure and a clear methodology. The Use scientifically tested programs
Author Response
Thanks the respectable reviewer for his/her valuable comment.
Reviewer 3 Report
The article is well written and the content and general logic of the manuscript is very accurate. The introductory section provides an adequate number of references. The conclusion section summarizes well the results of the field test. I suggest to revise the article taking into account of the following minor issues:
minor spell-check required:
line 70, "CO2" with "2" as subscript
lines 89-90, it would be useful to avoid repetitions ("experiment" and "experimental")
starting from eq. 20, please revise the numbering. It could be 16.
It might be useful to provide a new, more comprehensible, version of Figure 1.
Author Response
His article is well written and the content and general logic of the manuscript is very accurate. The introductory section provides an adequate number of references. The conclusion section summarizes well the results of the field test. I suggest to revise the article taking into account of the following minor issues: Response: Thanks the respectable reviewer for his/her valuable comment. minor spell-check required: line 70, "CO2" with "2" as subscript Response: Thank you for your attention, we have revised. For the modified part, we marked with red font line 70. lines 89-90, it would be useful to avoid repetitions ("experiment" and "experimental") Response: Thank you for your attention, we have revised. For the modified part, we marked with red font line 89-90. starting from eq. 20, please revise the numbering. It could be 16. Â Response: Thank you for your attention, we have revised. It might be useful to provide a new, more comprehensible, version of Figure 1. Response: Thanks the respectable reviewer for his/her valuable comment. We add figure 1. Backplane heat pipe air conditioning system. For the modified part, see page 5.